# Pain Pressure Threshold as a Non-Linear Marker of Neural Adaptation in Amputees: Evidence from the DEFINE Cohort

**DOI:** 10.3390/neurosci6010017

**Published:** 2025-02-18

**Authors:** Marta Imamura, Anna Carolyna Gianlorenço, Guilherme J. M. Lacerda, Linamara Rizzo Battistella, Felipe Fregni

**Affiliations:** 1Instituto de Medicina Física e Reabilitação, Hospital das Clínicas HCFMUSP, Faculdade de Medicina, Universidade de São Paulo, São Paulo 04116-040, Brazil; marta.imamura@fm.usp.br (M.I.); guilherme.lacerda@hc.fm.usp.br (G.J.M.L.); linamara@usp.br (L.R.B.); 2Neuromodulation Center and Center for Clinical Research Learning, Spaulding Rehabilitation Hospital and Massachusetts General Hospital, Harvard Medical School, Boston, MA 02138, USA; alepesteurgianlorenco@mgh.harvard.edu; 3Laboratory of Neuroscience and Neurological Rehabilitation, Physical Therapy Department, Federal University of Sao Carlos, Sao Carlos 13565-905, Brazil; 4Departamento de Medicina Legal, Bioética, Medicina do Trabalho e Medicina Física e Reabilitação do da Faculdade de Medicina da Universidade de São Paulo (FMUSP), São Paulo 01246-903, Brazil

**Keywords:** Pain Pressure Threshold (PPT), amputation, chronic pain, nonlinear associations, pain modulation

## Abstract

Background: Amputation poses significant physical, psychological, and emotional challenges, with chronic pain being one of the most debilitating outcomes. Pain Pressure Threshold (PPT), a measure of nociceptive sensitivity, is a valuable tool for assessing changes in pain perception. Understanding PPT modulation in amputees is crucial for uncovering the mechanisms underlying pain and developing targeted interventions for pain management. Objective: This study aimed to evaluate PPT in amputees and identify factors associated with PPT variation in this population. Methods: This cross-sectional study analyzed neurophysiological, clinical, and demographic data from 86 amputee patients. PPT was assessed as the primary outcome, and its associations with demographic and clinical predictors were examined using both linear and quadratic regression models. Results: Multivariate analysis identified a significant association between PPT and biological sex, with females exhibiting lower PPT values than males. Quadratic regression analyses revealed inverted U-shaped associations between PPT and age, BMI, and duration since amputation. PPT increased with age, peaking at 45.8 years, followed by a decline. Similar patterns were observed for BMI, peaking at 27.0 kg/m^2^, and for amputation duration, peaking at 26.6 months. Conclusions: Our findings indicate that sex, age, BMI, and time since amputation are significant factors influencing PPT in amputees, with nonlinear relationships observed for age, BMI, and amputation duration. These results suggest that physiological and disease-related factors (such as age, BMI, and duration of injury) have specific peaks for optimal PPT, highlighting their role in the brain’s compensatory system and potential implications for targeted pain management strategies.

## 1. Introduction

Amputation is defined as the loss or surgical removal of a body part, and in recent decades, it has contributed significantly to the global healthcare burden, resulting in disability and impairing mobility and physical function [1]. While the primary etiologies of amputation vary by region, diabetes and peripheral vascular disease have become the leading causes in high-income countries [2,3], while trauma remains the predominant cause of amputation in low-income countries [4]. Continuing pain and phantom limb pain phenomena (PLP) can significantly complicate recovery following amputation. These pain conditions arise from various complex physiological and neurological mechanisms. Studies regarding key predictors of post-amputation pain include pre-existing pain conditions, where individuals with chronic or pre-amputation pain are more likely to experience residual pain afterward [5,6]; psychological factors, such as higher levels of anxiety and depression, which increase the risk of chronic pain [7]; and the type and level of amputation [5]. Despite that, the mechanisms underlying pain after amputation remain incompletely understood, though they are believed to involve a combination of peripheral factors, such as neuromas, and central factors, including cortical reorganization and neuronal hyperexcitability.

Pain Pressure Threshold (PPT) is an important measure of nociceptive sensitivity, which quantifies the minimum pressure required to induce a painful sensation and relies on the function of sensory and nociceptive pathways in the nervous system, making it a reflection of both peripheral and central neurophysiological processes [8]. In knee osteoarthritis (KOA), lower PPTs are linked to functional outcomes like performance on the 10-Meter Walking Test and activity-related pain intensity; hence, greater disability corresponds with reduced pain thresholds [9]. Furthermore, patients with osteoarthritis-related pain who were scheduled for total knee replacement exhibited nervous system-origin hyperalgesia, which adversely affected pain levels, knee function, and various aspects of quality of life [10]. A study on fibromyalgia (FM) demonstrated that disinhibition of the descending pain modulatory system (DPMS) is positively correlated with dysfunction in peripheral sensory neurons, as assessed by quantitative sensory testing (QST), and is inversely associated with serum levels of brain-derived neurotrophic factor (BDNF), highlighting the complex interplay between central and peripheral mechanisms in chronic pain [11]. Additionally, the observation of movements of the left hand has been found to significantly reduce pain perception in that hand, an effect associated with decreased intracortical inhibition in the primary motor cortex. These findings underscore the link between motor control and pain processing and should inform the development of targeted therapies for pain management. Future research should explore combination therapies in patients with chronic pain to enhance treatment efficacy [12].

Few studies have investigated PPT in individuals after amputation. For example, Fuchs et al. recently assessed various sensory thresholds, including heat pain threshold (HPT), PPT, warmth detection threshold, and two-point discrimination, across multiple sites. Their results revealed no significant differences in any of the thresholds between the amputee group and healthy controls, suggesting that sensory perception was comparable across the groups studied [13]. However, other research has indicated that PPT may be altered at both the residual limb and other body sites following amputation [14]. The underlying neurophysiological mechanisms responsible for these alterations are thought to involve changes in peripheral nociceptive input, central sensitization, and maladaptive cortical reorganization. For instance, reduced PPT at the residual limb may reflect hyperalgesia or allodynia, often associated with post-amputation pain syndromes such as phantom limb pain or stump pain. Additionally, heightened sensitivity in some amputees may result from cross-modal plasticity, in which the central nervous system compensates for sensory loss by reorganizing somatosensory processing [15].

Understanding the modulation of PPT in amputees is critical for developing more effective pain management strategies, particularly in the context of phantom limb pain, residual limb pain, and other neuropathic conditions that are prevalent in this population. Based on that, our study aims to address this gap by examining both neurophysiological and clinical key factors of pain sensitization, including amputee patients with and without chronic pain.

## 2. Methods

### 2.1. Study Design

We performed a cross-sectional analysis of patients with amputations as part of an ongoing prospective cohort study titled “Deficit of Inhibition as a Marker of Neuroplasticity” (DEFINE study) in rehabilitation (ref). For this analysis, we utilized baseline data. The DEFINE study protocol received approval from the Research and Ethical Committee of Hospital das Clínicas at the University of São Paulo School of Medicine (HC FMUSP) under registration number 86832518.7.0000.0068.

### 2.2. Participants

The sample size consisted of 86 patients with lower limb amputation, which allowed for the modeling of up to 8 covariates and resulted in an effect size of approximately 0.3. The participants were included according to the following criteria:Inclusion Criteria: Participants were required to meet the following criteria: (1) adults aged 18 or older, (2) able to provide informed consent, (3) having taken stable medication for at least 3 months before the beginning of the study.Exclusion Criteria: Participants were excluded if they (1) had any unstable medical or social condition that would prevent them from participating, or (3) a history of neurological condition, such as traumatic brain injury, stroke, or spinal cord injury.

### 2.3. Study Procedures

Patients with amputations enrolled in the IMREA conventional rehabilitation program were invited to participate in the study and were included after providing informed consent. A qualified researcher conducted a range of clinical and neurophysiological assessments during a single visit. The selected assessment tools were designed to enable a thorough evaluation of the patients. All evaluations were performed by the same examiner, who was trained to ensure standardized procedures. One specific assessment, the Pressure Pain Threshold (PPT), was designated as the dependent variable for this study, aiming to enhance our understanding of pain sensitivity in this population.

### 2.4. Neurophysiological Assessment

#### 2.4.1. Pressure Pain Threshold (PPT)

PPT was defined as the lowest pressure level that causes pain, as measured with an algometer and recorded in kilopascals (kPa) and relies on the function of sensory and nociceptive pathways in the nervous system, making it a reflection of both peripheral and central neurophysiological processes [8]. We took three readings with 15 s intervals between each, and the average of these readings was used as the final PPT value. Measurements were taken in the thenar region on the right hand and on the left hand. The PPT bilateral was the average of the PPT left and right.

#### 2.4.2. Conditioned Pain Modulation (CPM)

For conditioned pain modulation (CPM), a protocol based on changes in PPTs was used [16]. Participants were asked to immerse one hand in a container of cold water (10–12 °C) for 1 min. Then, three algometry measures (with intervals of 15 s) on the contralateral hand were taken, and then, the average of these measurements was recorded. After an interval of 10 min (time for the body temperature to return to normal), the other hand was immersed in the cold water and followed the same protocol aforementioned. The CPM response was calculated using the difference between the average PPTs during the conditioned stimulus minus and the average PPTs. Positive values were related to an effective CPM, and negative values were associated with an ineffective CPM.

### 2.5. Clinical and Functional Variables

Demographic and baseline clinical information was extracted through a standardized medical interview, incorporating details about age, gender, race, marital status, body mass index, employment status, alcohol or smoking consumption, chronic health conditions, and long-term medications. The patient’s history of amputation was collected, such as time of lesion, time of hospitalization, etiology of the lesion, level of amputation, and presence and characteristics of pain and phantom sensations.

#### Clinical Scales

All variables collected in this study, including their detailed descriptions, measurement methods, and rationale for inclusion, can be found in the initial study protocol [17]. For this analysis, the clinical variables chosen by biological plausibility included pain VAS, Hospital Anxiety and Depression Scale (HADS) [18], Functional Independence Measure (FIM) [19], and The Montreal Cognitive Assessment (MOCA).

### 2.6. Statistical Analysis

We analyzed the variables and identified outliers in PPT, BMI, and amputation duration that could potentially skew results and affect reliability. Data points exceeding ±2 standard deviations were removed, and BMI values above 35 kg/m^2^ were considered outliers and were excluded from analysis. After examining the plotted variables, with a focus on assessing the outcome variable (PPT) in relation to potential predictors, we performed multivariate linear analyses on variables with potential linear associations. For variables displaying an inverted U-shape pattern, we conducted quadratic regression analyses. Clinically relevant variables were incorporated into each multivariate model using a stepwise approach to construct the final models, all of which were adjusted for sex, conditioned pain modulation (CPM), and depression. Regarding possible confounders, we analyzed them individually in separate models to assess their independent contributions while minimizing collinearity. Due to the exploratory nature of this study, we did not apply corrections for multiple comparisons, and we used only complete cases, as in our previous articles. While this approach may reduce power, we prioritized maintaining the dataset’s integrity and minimizing potential biases associated with imputation methods. It is worth mentioning that we tested additional variables, including education, pain intensity, pain duration, depression, Quebec Back Pain Disability Scale, FIM, anxiety, catastrophizing, and sleepiness, but none showed a significant association with PPT. *p*-values below 0.05 were considered statistically significant, and all statistical analyses were conducted using RStudio (Version 2023.06.0 + 421).

## 3. Results

### 3.1. Sample Characteristics

A total of 86 individuals participated in the study, with a mean age of 47.0 years (SD = 15.7). The sample was predominantly male (62.8%), while females comprised 12.8%. Regarding racial distribution, 27.9% of participants identified as White, and 47.7% as non-White. The mean body mass index (BMI) was 26.2 (SD = 5.81). The majority of amputations were nontraumatic in origin (57.0%), while 40.7% were classified as traumatic, and regarding the amputation level, 51.5% were above-knee, and 40.5% were below-knee amputations.

Conditioned pain modulation (CPM) values averaged 1.98 kPa (SD = 2.65). The mean duration since amputation was 23.7 months (SD = 17.9). Psychological assessments included a mean Hospital Anxiety Scale score of 4.12 (SD = 3.29) and a mean Hospital Depression Scale score of 2.68 (SD = 3.18). Functional independence, measured by the Functional Independence Measure (FIM), averaged 116.2 (SD = 5.0), and cognitive function, assessed using the Montreal Cognitive Assessment (MOCA), had a mean score of 21.75 (SD = 3.54). These baseline demographic and clinical characteristics are summarized in Table 1.

### 3.2. Outcome Characteristics

The primary outcome, Pain Pressure Threshold (PPT) measurements, were recorded for both the amputated and non-amputated sides. For the amputated side, the mean PPT was 8.48 kPa (95% CI = 7.89 to 9.07), with a standard deviation of 2.44. Similarly, the mean PPT for the non-amputated side was 8.73 kPa (95% CI = 8.1 to 9.36), with a standard deviation of 2.61. Both measurements had missing data in 21 cases, accounting for 24.1% of the sample. This is outlined in Table 2.

### 3.3. Multivariate Analysis

After a close examination of the plotted variables, specifically evaluating the outcome variable PPT against potential predictors, we conducted a multivariate linear analysis on variables that demonstrated a potential linear correlation. Among these, biological sex showed a statistically significant association with PPT (β = −2.13, 95% CI = −3.58 to −0.67, *p* < 0.01), suggesting that females have lower PPT than males. Subsequently, we conducted quadratic regression analyses for variables that exhibited an inverted U-shape association with PPT, resulting in three models. Age, including both its linear (β = 0.30, 95% CI = 0.14 to 0.47, *p* < 0.01) and quadratic terms (β = −0.003, 95% CI = −0.005 to −0.002, *p* < 0.01), demonstrated statistically significant associations with PPT, indicating an inverted U-shape relationship, with a positive association up to the vertex at 45.8 years and a negative association beyond that point. A similar inverted U-shape pattern was observed for BMI, including both its linear (β = 1.49, 95% CI = 0.22 to 2.76, *p* = 0.027) and quadratic terms (β = −0.03, 95% CI = −0.05 to −0.003, *p* = 0.036), with a positive association up to the vertex at 27.0 kg/m^2^ and a negative association afterward. Likewise, amputation duration, including its linear (β = 0.29, 95% CI = 0.14 to 0.44, *p* < 0.01) and quadratic terms (β = −0.005, 95% CI = −0.008 to −0.002, *p* < 0.01), demonstrated this pattern, showing a positive relationship up to the vertex at 26.6 months, followed by a negative association. The models are summarized in Table 3.

## 4. Discussion

In this study, we analyzed relationships between PPT and various sociodemographic and clinical variables using a combination of linear and quadratic regression models. Our primary findings indicate significant associations between PPT and key variables, highlighting both linear and nonlinear relationships. Biological sex showed a statistically significant linear association with PPT, with females exhibiting lower PPT values than males. Additionally, three variables—age, BMI, and amputation duration—demonstrated nonlinear associations with PPT, each following an inverted U-shape pattern. Specifically, age was positively associated with PPT up to a peak around mid-adulthood, after which the association turned negative. A similar inverted U-shape relationship was observed for BMI, indicating that PPT increased with BMI up to an optimal range before declining. Amputation duration followed this trend as well, with PPT increasing during the initial months post-amputation, reaching a peak, and subsequently decreasing.

### 4.1. Association Between PPT and Biological Sex

Biological sex was found to have a statistically significant association with PPT, with females exhibiting lower PPT values than males. This negative association suggests a sex-related difference in sensory perception, where females may experience heightened sensitivity to pressure pain. These findings align with the existing literature, showing that women present increased sensitivity to pressure and heat pain stimuli in other conditions such as neuropathic pain, knee osteoarthritis, and migraine. For example, in the study conducted by Bartley et al. (2015) [20], women with knee osteoarthritis exhibited greater sensitivity to various nociceptive stimuli compared to their male counterparts. Specifically, women demonstrated lower pain thresholds and tolerances to heat, cold, and pressure stimuli, as well as increased sensitivity to mechanical pressure pain and higher pain scores in response to mechanical and thermal stimuli. However, in contrast to the observed sex differences in experimental pain responses, no significant differences were found between men and women on clinical pain measures, except for women reporting more widespread pain [20]. Studies incorporating female participants have highlighted significant sex differences in the physiological mechanisms underlying pain. These differences include sex-specific roles of genes and proteins, as well as distinct interactions between hormones and the immune system that modulate pain signal transmission [21]. Neuroimaging studies have further revealed sex and gender differences in the neural circuitry involved in pain processing, with distinct brain alterations observed in chronic pain conditions between men and women. Clinical pain research also indicates that gender can influence how individuals perceive, contextualize, and cope with pain, potentially affecting the experience of pain and the development of chronic pain. Additionally, sex and gender biases in clinical settings may impact both the perception and treatment of pain, contributing to disparities in pain management outcomes [22,23].

### 4.2. Association Between PPT and Biological Age

Our analysis revealed an inverted U-shape relationship between PPT and age, with a positive association up to a peak at 45.8 years (Figure 1), followed by a negative association thereafter. To our knowledge, this is the first study to report this pattern. In relation to the existing literature, a systematic review and meta-analysis, which examined the influence of age on PPT in healthy participants, suggested that PPT was lower in old adults compared with younger adults (*p* = 0.018, *I*^2^ = 60.970%) [24]. The literature also shows that the prevalence of pain varies by age and sex across different pain conditions [25]. Certain pain conditions, such as migraine headaches [26] and temporomandibular disorder [27], show a decline in prevalence after the fourth decade of life [28], suggesting a potential age-related shift in pain perception around this period. Given these findings, our observed inverted U-shape relationship between age and PPT may reflect broader changes that influence pain sensitivity over time. The initial increase in PPT up to mid-adulthood could indicate more robust pain modulation mechanisms in younger adults, while the subsequent decline in older age might correlate with factors such as increased incidence of chronic pain conditions, age-related changes in sensory processing, or central sensitization. Together, these findings emphasize the complex and potentially dynamic influence of age on pain sensitivity, suggesting that future research should further explore age-related mechanisms that contribute to changes in PPT across the lifespan.

### 4.3. Association Between PPT and Biological BMI

A similar inverted U-shape pattern emerged between BMI and PPT. This association revealed a positive relationship up to a peak BMI of 27.0 kg/m^2^ (Figure 2), after which the relationship turned negative. This finding suggests that within a certain range, higher BMI may be associated with increased PPT, potentially indicating enhanced pain tolerance or reduced sensitivity to pressure pain. Beyond this threshold, however, higher BMI is associated with decreased PPT, possibly due to changes such as inflammation and altered pain processing mechanisms, which are often linked to obesity [29].

In fact, a previous study has shown that, once BMI exceeds a certain threshold, it can act as an effect modifier in fibromyalgia patients, disrupting pain compensation mechanisms and impairing the system’s overall functionality [30]. Another study suggests that obesity disrupts pain mechanisms and their compensatory effects, playing a role in exacerbating fibromyalgia symptoms and impairing physiological pain-inhibitory mechanisms [31]. Given these findings, our observed inverted U-shape relationship between BMI and PPT highlights the complex role BMI plays in pain perception. The initial increase in PPT up to a BMI of 27.0 kg/m^2^ may reflect a range where a higher BMI corresponds to increased pain tolerance or reduced sensitivity to pressure pain. However, as BMI continues to rise beyond this threshold, the decline in PPT could be linked to physiological changes commonly associated with obesity, such as inflammation, altered pain processing, and disrupted pain-inhibitory mechanisms. Together, these findings underscore the dynamic influence of BMI on pain sensitivity, suggesting that pain perception may vary significantly across different BMI ranges.

### 4.4. Association Between PPT and Time Since Amputation

With respect to amputation-related factors, our analysis revealed that time since amputation exhibited an inverted U-shape relationship with PPT. Specifically, there was a positive correlation with time since amputation up to a peak at approximately 26.6 months (Figure 3), after which the relationship became negative. This trend may reflect a complex interplay of factors, including the resolution of acute post-surgical pain, the development of chronic pain conditions such as phantom limb pain, and the potential for cortical reorganization or central sensitization. For instance, in the early months following amputation, pain sensitivity may be heightened due to inflammation, nerve injury, or maladaptive central pain processing. As time progresses, there may be a reduction in pain sensitivity as peripheral healing occurs and the individual adapts to the loss of the limb. However, after a certain point, further increases in pain sensitivity may occur due to long-term complications such as residual limb pain, phantom limb pain, or central nervous system changes, including central sensitization. Thus, time since amputation appears to play a critical role in modulating pain perception, with variations in PPT potentially reflecting both peripheral and central pain mechanisms that evolve over time.

## 5. Limitations

Given its exploratory design, our study has several limitations. First, the lack of a control group restricts the generalizability of our findings to amputee patients. Additionally, we did not apply corrections for multiple comparisons, which may have increased the risk of Type I errors. However, despite these limitations, our sample size was adequate to detect significant associations with up to eight covariates. Nevertheless, these findings should be interpreted with caution, and further validation in larger, more rigorous studies is needed to confirm the results.

## 6. Conclusions

In conclusion, our findings underscore a significant association between sex and PPT, with female amputees exhibiting lower PPT values compared to male amputees. Additionally, multivariate analyses revealed a nonlinear, inverted U-shaped relationship between PPT and key factors such as age, BMI, and time since amputation. These results suggest that both physiological factors and temporal variables play crucial roles in modulating nociceptive sensitivity in amputee patients.

Furthermore, our findings provide preliminary insights into pain management for amputees, emphasizing the role of endogenous pain modulation. Pharmacological approaches may benefit from personalized treatments targeting individual pain modulation differences. Physiotherapeutically, tailored rehabilitation strategies, including graded motor imagery, desensitization techniques, neurostimulation (e.g., TMS, tDCS), and exercise programs enhancing endogenous pain control, could help restore pain modulation and reduce chronic pain symptoms. These findings support a multi-modal approach to pain management in this population.

## Figures and Tables

**Figure 1 neurosci-06-00017-f001:**
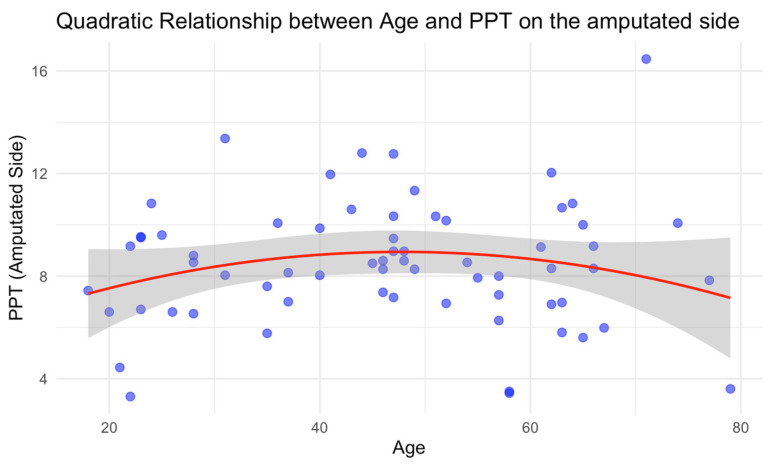
Quadratic relationship between BMI and Pain Pressure Threshold (PPT) on the amputated side, showing a peak around 46 years before declining. The shaded region represents the confidence interval.

**Figure 2 neurosci-06-00017-f002:**
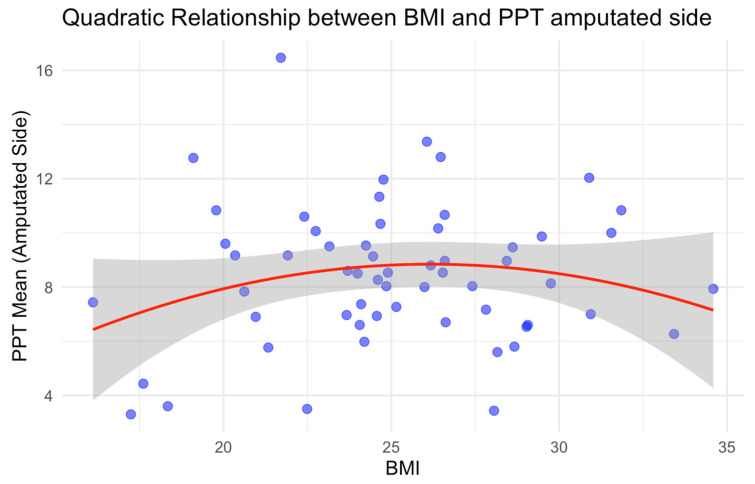
Quadratic relationship between BMI and Pain Pressure Threshold (PPT) on the amputated side, showing a peak around a BMI of 27 kg/m^2^ before declining. The shaded region represents the confidence interval.

**Figure 3 neurosci-06-00017-f003:**
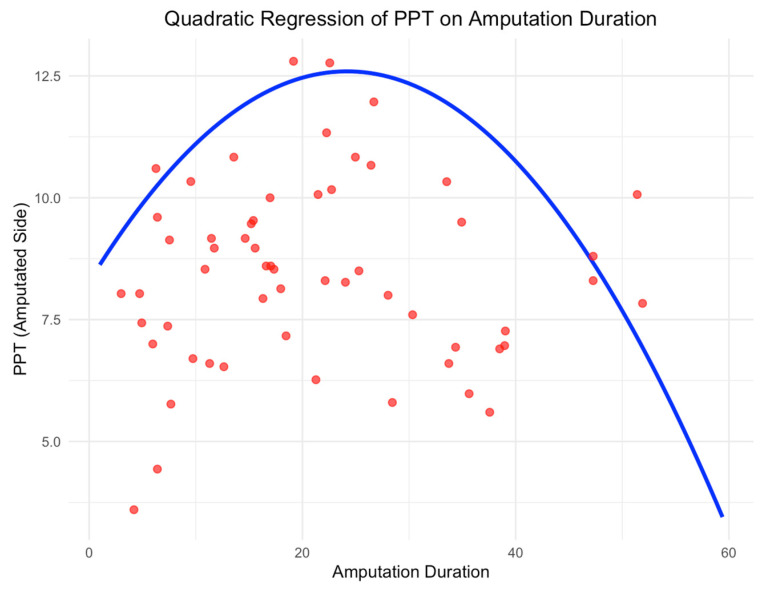
Quadratic regression of Pain Pressure Threshold (PPT) on amputation duration, showing a peak around 26.6 months before declining.

**Table 1 neurosci-06-00017-t001:** Baseline demographic and clinical characteristics of study participants (n = 86).

Variable	
Age (years)—mean (SD)	47.0 (15.7)
Gender—No. (%)	
Male	54 (62.8%)
Female	11 (12.8%)
Race—No. (%)	
White	24 (27.9%)
Non-White	41 (47.7%)
Body mass index (kg/m^2^)—mean (SD)	26.2 (5.81)
Etiology of lesion—No. (%)	
Traumatic	35 (40.7%)
Nontraumatic	49 (57.0%)
Conditioned pain modulation (kPa)—mean (SD)	1.98 (2.65)
Amputation duration (months)—mean (SD)	23.7 (17.9)
Hospital Anxiety Scale—mean (SD)	4.12 (3.29)
Hospital Depression Scale—mean (SD)	2.68 (3.18)
Functional Independence Measure—mean (SD)	116.2 (5.0)
Montreal Cognitive Assessment—mean (SD)	21.75 (3.54)

Values are presented as mean (SD) for continuous variables and percentages for categorical variables. No.: number/count; SD (Standard Deviation); kPa: Kilopascal.

**Table 2 neurosci-06-00017-t002:** Characteristics of dependent variables.

Variable	Mean (95% CI)	SD	Missing Data: Number (%)
PPT Amputated Side (kPa)			21 (24.1%)
Male	8.71 (7.99–9.42)	2.62	
Female	7.43 (6.43–8.43)	1.49	
PPT Non-Amputated Side (kPa)			21 (24.1%)
Male	8.90 (8.12–9.68)	2.84	
Female	8.12 (7.20–9.04)	1.36	

Pain Pressure Threshold (PPT). CI: confidence interval; SD (standard deviation); kPa: kilopascal.

**Table 3 neurosci-06-00017-t003:** Multivariable model of PPT amputated side.

Variable	β-Coefficient (95%CI)	*p*-Value	Adjusted R^2^
Model 1: Sex	−2.13 (−3.58 to −0.67)	<0.01	0.22
Model 2: Age			0.36
Quadratic Term	−0.003 (−0.005 to −0.002)	<0.01	
Linear Term	0.30 (0.14 to 0.47)	<0.01	
Model 3: BMI			0.28
Quadratic Term	−0.03 (−0.05 to −0.003)	0.036	
Linear Term	1.49 (0.22 to 2.76)	0.027	
Model 4: Amputation duration			0.35
Quadratic Term	−0.005 (−0.008 to −0.002)	<0.01	
Linear Term	0.29 (0.14 to 0.44)	<0.01	

Adjusted by sex, CPM, and depression; BMI: body mass index.

## Data Availability

The data supporting the findings of this study are available from the corresponding author upon reasonable request due to privacy restrictions.

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
