# Peer review of "Pain Pressure Threshold as a Non-Linear Marker of Neural Adaptation in Amputees: Evidence from the DEFINE Cohort"

_neurosci, 2025, doi:10.3390/neurosci6010017_

Round 1

Reviewer 1 Report

Comments and Suggestions for Authors

In this study, authors tried to evaluate PPT in amputees and identify factors associated with PPT variation in this population. They concluded, well specifying study limitations, that sex, age, BMI, and time since amputation are significant factors influencing PPT in amputees, with non-linear relationships observed for age, BMI, and amputation duration; physiological and disease-related factors have specific peaks for optimal PPT, highlighting their role in the brain's compensatory system and potential implications for targeted pain management strategies.

Study was conducted with good scientific method but I think it needs some major revisions:

1) Abstract: no comments

2) Introduction: no comments

3) Methods: it would be useful if the authors explained well which type of amputation they considered (upper limb or lower limb or indifferent) and the percentage of corresponding patients (how many patients with upper or lower limb amputation); Part 2.5.1 "Clinical scales" should be remodeled to avoid explaining in detail the various scales used (the reader should already know them and know their meaning)

4) Results: captions should be added to the tables to help the reader understand their meaning

5) Discussion, Limitation, Conclusion: no comments

6) References: should be improved considering that more than 50% (18/28) of the references are more than 5 years old

7) minor revision of English

Comments on the Quality of English Language

Minor revision of English

Author Response

Reviewer 1

In this study, authors tried to evaluate PPT in amputees and identify factors associated with PPT variation in this population. They concluded, well specifying study limitations, that sex, age, BMI, and time since amputation are significant factors influencing PPT in amputees, with non-linear relationships observed for age, BMI, and amputation duration; physiological and disease-related factors have specific peaks for optimal PPT, highlighting their role in the brain's compensatory system and potential implications for targeted pain management strategies.

Study was conducted with good scientific method but I think it needs some major revisions:

1) Abstract: no comments

2) Introduction: no comments

3) Methods: it would be useful if the authors explained well which type of amputation they considered (upper limb or lower limb or indifferent) and the percentage of corresponding patients (how many patients with upper or lower limb amputation); Part 2.5.1 "Clinical scales" should be remodeled to avoid explaining in detail the various scales used (the reader should already know them and know their meaning)

Response: Thank you for your comment. All participants in our study had lower limb amputations, which we have clarified in the Methods section. Additionally, the percentage distribution of amputation levels is provided in the Results section. Regarding Section 2.5.1 on clinical scales, we have made the necessary adaptations as suggested.

4) Results: captions should be added to the tables to help the reader understand their meaning

Response: Thank you for your suggestion. We added captions to the tables.

5) Discussion, Limitation, Conclusion: no comments

6) References: should be improved considering that more than 50% (18/28) of the references are more than 5 years old

Response: Thank you for your suggestion. We have added more updated references to improve the manuscript.

7) minor revision of English

Response: We revised it. Thank you for your feedback.

Reviewer 2 Report

Comments and Suggestions for Authors

This is a study of experimentally tested pressure pain sensitivity in amputees with some interesting results. It is well-written and has a good background. I have a few remarks that need to be considered:

-More descriptives are required about the study population. Do they all comprise lower limb amputees? If not, how many are upper limb, how many lower limb amputees? How many above knee, below knee etc?, number of amputees per non-traumatic etiology (e.g. diabetes, vascular), unilateral or bilateral amputation, amputation due to chronic or acute disease (acute thrombosis, trauma), use of prosthetics or not, use of medications

- Measurements of PPT were conducted over the thenar muscles. As one might expect measurements to be made on the residual limb, the rationale for the location should be explained.

-How did authors handle statistically the missing data? Were the cases completely excluded from the analysis? This leads to reduced power.

-Why should the regression model be adjusted for CPM? This might have mitigated correlations with other clinical predictors. Lowered PPT and impaired CPM may share the same pathophysiological substrates, and CPM would be as interesting as PPT as a dependent valuable in this study.

-I was impressed by the low mean MOCA Test score of 21.75 for a mean 47 yrs of age of the study population indicating that many of the amputees studied fall within the spectrum of mild cognitive impairment. I wonder whether this could be at least in part effect of the medications received and would be interested to know the medication status of the study participants (proportion of patients using opioids, tranquilisers and anticonvulsants like pregabalin) which could also be included in the regression analysis to find out if medications influence the primary outcome.

-To incorporate a categorical value as sex in the regression analysis, dummy coding must have been used. Authors report a statistically significant association between biological sex and PPT. The β coefficient is -2.13 which makes little sense, in terms of the notion that the beta coefficient represents the estimated change in the dependent variable for a one-unit change in a predictor variable. So it is advisable to also report mean values of PTT for males and females.

-It is advisable to include plots in the paper to graphically illustrate the findings of the study, the U-shaped relations in particular.  

-It is important to mention which candidate variables were tested in the regression models even if they were not shown as significant predictors. For example clinical phantom limb or residual limb pain intensity, level of depression and anxiety symptoms, underlying cause of amputation (traumatic or non-traumatic) etc, and explicitly state that factors a, b, c were not associated with the outcome of PPT.    

-Did the authors check multiple predictor variables simultaneously in the same regression model or rather check each candidate predictor variable separately, only adjusted for sex, CPM and depression as mentioned? For example, including age and duration of amputation as predictors in the same model might have revealed collinearity issue, and excluded one of the two variables as having secondary association with PPT.

-How could the results of this study, although preliminary, impact on pain management of these patients, both pharmaceutically and physiotherapeutically? 

Author Response

Reviewer 2

This is a study of experimentally tested pressure pain sensitivity in amputees with some interesting results. It is well-written and has a good background. I have a few remarks that need to be considered:

-More descriptives are required about the study population. Do they all comprise lower limb amputees? If not, how many are upper limb, how many lower limb amputees? How many above knee, below knee etc?, number of amputees per non-traumatic etiology (e.g. diabetes, vascular), unilateral or bilateral amputation, amputation due to chronic or acute disease (acute thrombosis, trauma), use of prosthetics or not, use of medications

Response: Thank you for your feedback. We have updated the manuscript to include information confirming that all patients are lower limb amputees, along with details on their characteristics.

- Measurements of PPT were conducted over the thenar muscles. As one might expect measurements to be made on the residual limb, the rationale for the location should be explained.

Response: As mentioned above, all patients had lower limb amputations; therefore, PPT measurements were conducted on the upper limb (thenar muscles). This approach ensures consistency and avoids potential variability associated with residual limb conditions.

-How did authors handle statistically the missing data? Were the cases completely excluded from the analysis? This leads to reduced power.

Response: We used only complete cases, as we have done in most of our previous articles. Although this approach reduces power, we chose it to maintain the integrity of the dataset and avoid potential biases introduced by imputation methods. Given the exploratory nature of our study, imputing missing values could have led to inaccuracies in key variables, particularly those related to pain perception and neurophysiological measures.

-Why should the regression model be adjusted for CPM? This might have mitigated correlations with other clinical predictors. Lowered PPT and impaired CPM may share the same pathophysiological substrates, and CPM would be as interesting as PPT as a dependent valuable in this study.

Response: We adjusted the regression model for CPM because it is a key indicator of endogenous pain modulation and could influence the relationship between other clinical variables and PPT. Given that lowered PPT and impaired CPM may share common pathophysiological mechanisms, adjusting for CPM helps isolate the independent effects of other predictors on PPT. While CPM itself could indeed be an interesting dependent variable, our primary focus in this study was on PPT, as it provides a direct measure of mechanical pain sensitivity in this population.

-I was impressed by the low mean MOCA Test score of 21.75 for a mean 47 yrs of age of the study population indicating that many of the amputees studied fall within the spectrum of mild cognitive impairment. I wonder whether this could be at least in part effect of the medications received and would be interested to know the medication status of the study participants (proportion of patients using opioids, tranquilisers and anticonvulsants like pregabalin) which could also be included in the regression analysis to find out if medications influence the primary outcome.

Response: Thank you for your comment. After reviewing our records, we found that only a small proportion of participants were taking these medications. For instance, less than 20% of patients were using gabapentin, and the use of opioids, tranquilizers, and other anticonvulsants was also relatively low. Given this limited exposure, we opted not to include medication use as a covariate in the regression analysis, as its statistical influence on the primary outcome would likely be minimal. However, we acknowledge the potential impact of medications on cognitive function and pain perception and have noted this as a consideration for future studies with larger sample sizes and more balanced medication distributions.

-To incorporate a categorical value as sex in the regression analysis, dummy coding must have been used. Authors report a statistically significant association between biological sex and PPT. The β coefficient is -2.13 which makes little sense, in terms of the notion that the beta coefficient represents the estimated change in the dependent variable for a one-unit change in a predictor variable. So it is advisable to also report mean values of PTT for males and females.

Response: Thank you for your comment. We confirm that dummy coding was applied, with males coded as 0 and females coded as 1. The β coefficient of -2.13 indicates that, on average, females have 2.13 units lower PPT than males, which aligns with our expectations given known sex differences in pain sensitivity. To provide additional clarity, we have now reported the mean PPT values for males and females in the results section.

-It is advisable to include plots in the paper to graphically illustrate the findings of the study, the U-shaped relations in particular.  

Response: Absolutely. We inserted plots in the manuscript.

-It is important to mention which candidate variables were tested in the regression models even if they were not shown as significant predictors. For example clinical phantom limb or residual limb pain intensity, level of depression and anxiety symptoms, underlying cause of amputation (traumatic or non-traumatic) etc, and explicitly state that factors a, b, c were not associated with the outcome of PPT.  

Response: Thank you for your comment. In addition to the variables reported in the manuscript, we also tested education, pain intensity, pain duration, depression, Quebec Back Pain Disability Scale, FIM, anxiety, catastrophizing, and sleepiness in the regression models. However, these factors were not significantly associated with PPT and were therefore not included in the final model. We will explicitly state this in the manuscript to clarify the range of candidate variables that were examined.

-Did the authors check multiple predictor variables simultaneously in the same regression model or rather check each candidate predictor variable separately, only adjusted for sex, CPM and depression as mentioned? For example, including age and duration of amputation as predictors in the same model might have revealed collinearity issue, and excluded one of the two variables as having secondary association with PPT.

 Response: Thank you for your comment. We checked the covariates one by one in separate models. We have now added this explanation to the manuscript to clarify our approach. This method allowed us to assess the independent contribution of each variable while minimizing potential collinearity issues.

-How could the results of this study, although preliminary, impact on pain management of these patients, both pharmaceutically and physiotherapeutically? 

Response: The findings of this study, despite being preliminary, offer important insights into pain management for amputees. Pharmaceutically, our results suggest that individual differences in pain perception may be linked to endogenous pain modulation mechanisms such as CPM. This highlights the potential need for a more personalized approach to pharmacological treatments. Physiotherapeutically, the observed differences in pain thresholds and modulation mechanisms suggest that tailored rehabilitation programs, such as graded motor imagery, desensitization techniques, and neurostimulation-based interventions (e.g., TMS or tDCS), could be beneficial in restoring pain modulation and reducing chronic pain symptoms. Additionally, optimizing exercise protocols that enhance endogenous pain control mechanisms may be an important avenue for future clinical interventions. We have briefly included this in the conclusion of the manuscript.

Round 2

Reviewer 1 Report

Comments and Suggestions for Authors

Authors accurately improved manuscript according to my comments.

It could be published in this form.

Reviewer 2 Report

Comments and Suggestions for Authors

Remarks have been satisfactorily addressed.